# Quantitative Determination of Bisphenol A and Its Congeners in Plant-Based Beverages by Liquid Chromatography Coupled to Tandem Mass Spectrometry

**DOI:** 10.3390/foods11233853

**Published:** 2022-11-29

**Authors:** Marica Erminia Schiano, Federica Sodano, Chiara Cassiano, Ferdinando Fiorino, Serenella Seccia, Maria Grazia Rimoli, Stefania Albrizio

**Affiliations:** 1Department of Pharmacy, University of Naples Federico II, 80131 Naples, Italy; 2Institute for Polymers, Composites and Biomaterials, Italian National Research Council, 80078 Naples, Italy; 3Interuniversity Consortium INBB, Viale delle Medaglie d’Oro, 305, 00136 Rome, Italy

**Keywords:** bisphenols, liquid chromatography with tandem mass spectrometry (LC-MS/MS), plant-based beverages, solid-phase extraction, risk assessment

## Abstract

The consumption of plant-based beverages as an alternative to cow’s milk has recently gained vast attention worldwide. The aim of this work is to monitor the intake of Bisphenol A (BPA), Bisphenol B (BPB) and Bisphenol S (BPS) in the Italian population through the consumption of these foodstuffs. Specifically, the development and validation of an analytical procedure for the quantitative determination of the analytes by liquid chromatography coupled to tandem mass spectrometry was reported. Thirty-four samples of plant-based beverages (soya, coconut, almond, oats and rice) of popular brands marketed in Italy were analyzed. BPA was found in 32% of the samples, while BPB was found in 3% of the samples. The risk assessment using the Rapid Assessment of Contaminant Exposure (RACE) tool demonstrated that there was no risk for all population groups, when using the current Tolerable Daily Intake (TDI) of 4 ng/kg body weight (bw)/day as a toxicological reference point. In contrast, using the new temporary TDI of 0.04 ng/kg bw/day, the existing risk was found to be real for all population groups. If this value were to become final, even more attention would have to be paid to the possible presence of BPA in food to protect consumer health.

## 1. Introduction

In recent years, the plant-based beverages market has been gaining in popularity compared to cow’s milk. The term plant-based beverage replaced the former name plant milk following European Commission Decision No. 791 of 20 December 2010. The new rule established that the designations ‘milk’, ‘cream or custard’, ‘butter’, ‘cheese’ and ‘yoghurt’ are restricted to products of animal origin—and consequently cannot legitimately be used to designate purely plant-based products, with clearly defined exceptions. The purpose of this rule was to avoid misleading consumers. Many reasons underlie this growing trend in consumer tastes. The first is undoubtedly for health reasons. In fact, an increasing number of people suffer from cow’s milk protein allergy or lactose intolerance and as a result, must eliminate cow’s milk from their diet. Many people choose to avoid consuming foods of animal origin for an ethical motivation. Some people simply tend to follow the trend of vegan or vegetarian diets, which are considered healthier than diets that include meat and foods of animal origin [1]. In addition, based on the amounts of greenhouse gases emitted, the environmental impact of producing plant-based beverages appears to be less than that of cow’s milk production [1,2,3]. According to the Veganok Observatory report published in 2020, the global market for plant-based drinks has been valued at about $12 billion in 2019 and is expected to grow to $22.4 billion by 2025 (https://www.vegansociety.com/news/market-insights/dairy-alternative-market/european-plant-milk-market (accessed on 22 August 2020); www.osservatorioveganok.com (accessed on 22 August 2020). Unfortunately, the potential food contamination by chemical compounds is one of the most serious food safety issues today. Therefore, the evaluation of the benefits of a food in the diet cannot be separated from the assessment of its level of contamination. Chemical contamination of food can occur by transfer of chemicals from the environment to the food chain and/or by migration from Food Contact Materials (FCMs). In the latter case, all stages of food manufacturing, from processing to transportation, can be a critical point for chemical contamination.

Recently, the attention of many researchers has been focused on a group of chemical compounds called Endocrine Disrupting Chemicals (EDCs) because of their property of interfering with the proper functioning of the endocrine system [4,5].

Bisphenol A (BPA) represents one of the best known and most studied members of the groups of EDCs. Its toxicity is also related to several pathologies such as metabolic, developmental and mental disorders, immune system impairments, and, more recently, microbiota alterations [6,7,8,9,10,11]. BPA is prevalently used as a plasticizer in the production of many plastic items and FCMs [12]. Because of the migration phenomenon, BPA can easily contaminate foods that come into contact with BPA-based plastic materials. As a result, European legislation has fixed a Specific Migration Limit (SML) for BPA at 0.05 mg/kg of food (EC Regulation No. 2018/213) to protect consumer health. In addition, the migration of BPA from paints and coatings applied to materials or articles intended for the packaging of foods for infants and young children is completely banned to better protect these groups most vulnerable to the toxic effects of BPA. Late last year, European Food Safety Authority (EFSA) recommended lowering the Tolerable Daily Intake (TDI) value from 4 µg/kg bw/day to 0.04 ng/kg bw/day in view of new scientific evidence on the effects of BPA on the immune system (https://www.efsa.europa.eu/en/news/bisphenol-efsa-draft-opinion-proposes-lowering-tolerable-daily-intake, accessed on 28 February 2022). To overcome restrictions on its use, some chemical analogs have been proposed to replace BPA in plastic materials. Unfortunately, due to their structural similarities, these compounds show a comparable toxicity to BPA [13,14,15,16,17]. To date, no legal limit has yet been set for BPA substitutes, with the exception of Bisphenol S (BPS) (SML, 0.05 mg/kg of food, EC Regulation No. 2018/213). In recent years, our research group has conducted many works aimed at tracking potential sources of population exposure to BPAs through the diet [18,19,20,21].

Considering their increasing use, plant-based beverages may represent a non-negligible source of BP in the human diet. Even though they are mainly packaged in tetra pack material consisting of paper, polyethylene and aluminum, BPs contamination could equally occur during food production steps [18,20,21,22]. Consistently, in the present study, thirty-four plant-based beverage samples were analyzed to monitor the presence of BPA, BPB, and BPS. The analysis of the present samples was preceded by the development and validation of an appropriate analytical procedure to quantify the analytes, consisting of Solid-Phase Extraction (SPE) and qualitative-quantitative analysis by Liquid Chromatography Electrospray Ionization triple-quadrupole Tandem Mass Spectrometry (LC-ESI-QqQ-MS/MS) of the samples. Subsequently, the validated method was used to analyze plant-based beverages of popular brands marketed in Italy to monitor their potential contamination by BPs. Finally, the Rapid Assessment of Contaminant Exposure (RACE) tool promoted by the European Food Safety Authority (EFSA) was applied to assess the risk to the Italian population related to BPs exposure through consumption of plant-based beverages.

## 2. Materials and Methods

### 2.1. Reagents and Chemicals

The standards used for validation of the analytical method, BPA (purity grade ≥ 99%), BPB (purity grade ≥ 98%) and BPS (purity grade ≥ 98%), were all purchased from Sigma-Aldrich (Milan, Italy). Methanol (MeOH) suitable for HPLC, grade ≥ 99.9%, was supplied by Carlo Erba (Milan, Italy). Ultrapure water was produced in the laboratory using Elix Essential Water Purification System (Merck Millipore, Burlington, MA, USA). Strata X-PRO (500 mg/6 mL) cartridges were bought from Phenomenex (Torrance, CA, USA). Stock solutions of individual BP standards were prepared by accurately weighing 5.0 ± 0.1 mg of each analyte, in dark glass vials, and dissolving them in 5 mL of MeOH. The standard solutions were stored at −20 °C for up to 4 months. Working standard solutions of BPA, BPB and BPS were prepared by combining aliquots of each stock solution and diluting them in MeOH to obtain a final concentration of 100 ng/mL. After preparation, the working solutions were stored at −20 °C and, before use, were kept at room temperature and vortexed for 1 min. Two series of calibration standards in the range of 0.78–50 ng/mL were prepared by serial dilution in MeOH and plant-based beverage extracts, respectively.

### 2.2. Real Samples

A total of 34 soya, oats, rice almond and coconut beverage samples were collected in the province of Naples, Italy, between January and March 2021 and analyzed as described above. Eleven different brands were selected. The samples covered five different tastes: soya (*n* = 9); oats (*n* = 8); rice (*n* = 8); almond (*n* = 7); coconut (*n* = 2). All samples were packaged in tetra pack; 27 in 1 L packages and 7 in 500 mL packages) [22]. The 34 samples were all stored at room temperature and analyzed before the expiration date.

### 2.3. Sample Preparation

#### 2.3.1. Sample Pretreatment

Each package of plant-based beverage was shaken manually to make the content homogeneous before sampling. Then, 2 mL of sample was taken, mixed with 4 mL of MeOH and placed in a glass tube. The remaining content of each package was frozen and stored at −20 °C for subsequent analysis. To make the solution homogeneous and to extract the analytes, a four-step procedure was applied: (1) vortex agitation for 1 min using a Vortex ZX4 shaker equipped with an infrared (IR) system from Hosmotic (Naples, Italy); (2) sonication in an ultrasonic bath for 15 min at room temperature using a Branson 2210R-MT Ultrasonic (Branson Ultrasonics Corp., Brookfield, CT, USA); (3) centrifugation at 2400 rpm for 15 min using an Allegra X-30R high volume centrifuge equipped with fixed angle rotors from Beckman Coulter (Brea, CA, USA); (4) finally, the supernatant was recovered. All steps from (1) to (4) were repeated two times and the supernatants were combined before the successive SPE step. The procedure of sample pretreatment is schematically illustrated in Figure 1.

#### 2.3.2. Sample Extraction Procedure by SPE

SPE was performed on 500 mg/6 mL Strata-X Pro cartridges (Torrance, CA, USA). Each cartridge was conditioned with 4.0 mL of MeOH followed by 4.0 mL of ultrapure water before loading the pretreated sample. The cartridge was washed with 4.0 mL of ultrapure water to remove impurities from the sample and then dried under a vacuum for 10 min. The analytes were eluted with 4.0 mL of MeOH under vacuum. The eluate obtained was dried by distillation in a rotary evaporator at 35 °C using a Rotavapor R-100 by Buchi (Milan, Italy). Subsequent quantitative analysis by LC-MS/MS was undergone on the residue reconstituted with 1.0 mL of ultrapure water/MeOH 50/50 *v*/*v* and vortexed for 1 min. Before each extraction process, the glassware and plastic equipment, used during the analyses, were thoroughly washed in MeOH to avoid any possible background contamination [23]. All solid-phase extraction operations were shown in Figure 2.

### 2.4. LC-ESI-QqQ-MS/MS Analysis

Analyses were conducted with an Agilent 6470 LC/ESI-TQ system equipped with a Jet Stream ion source operated in negative ion mode. Chromatographic separation was performed with an Agilent 1290 Series UHPLC (Santa Clara, CA, USA), equipped with a Luna Polar 1.7 µm, 100 Å, 50 mm × 2.1 mm stainless steel column (Phenomenex, Torrance, CA, USA). During the analysis, the flow rate was set at 0.400 mL/minute and the column temperature at 45 °C. A sample volume of 5 µL was injected.

Separation was achieved by a linear gradient from 0.01% acetic acid in ultrapure water to 0.01% acetic acid in MeOH as displayed in Table 1. The time for post-run column re-equilibration was fixed at 2 min. The mass spectrometer was periodically calibrated in the mass range 112.99–2833.87 amu with the reference standard mixture solutions provided by the manufacturer. Mass Hunter Workstation software (Agilent, Santa Clara, CA, USA) was used for data acquisition and processing. Analyses were conducted in multiple-reaction monitoring (MRM) mode.

Flow injections of individual standard solutions (BPA, BPB and BPS) at 1000.0 ng/mL were employed to optimize source parameters with LC flow conditions. The following experimental parameters were optimized: Gas temperature 200 °C, Gas flow: 11 L/min, Nebulizer: 45 psi, Sheat gas temperature 350 °C, Sheat gas flow: 12 L/min, Ion spray voltage −3500 V, Noozle voltage 2000 V. The MRM transition was optimized by acquiring the product ion spectra and using Optimizer software provided by the LC-MS manufacturer. For all BPs, tandem mass spectrometry analyses were performed in multiple reaction and negative ionization monitoring mode (-MRM). For each precursor ion (mass Q1), two product ions (masses Q3) were selected, one for quantification and the other for confirmation, identifying a quantifier ion (Q) and a qualifier ion (q). Both Q1 and Q3 were operated at unity resolution with a cell accelerator voltage of 7 V, and 150 ms was the dwell time allowed for each transition. Table 2 displays the selected quantitative and qualitative transitions for each analyte. Identification of the target compounds was based on comparison of the retention time (t_R_) of the chromatographic peaks of the quantifying and qualifying ions with the peaks of the reference standards.

### 2.5. Statistical Analysis

Data processing and statistical interpretation were performed with Microsoft Excel 2016. Experimental data were collected in triplicate, and experimental results are presented as mean values. A one-way ANOVA was performed using GraphPad Prism (GraphPad prism, version 8.4.3, Chicago, IL, USA) to explore potential significant disparities between the samples. The significance level was set at *p* = 0.05.

### 2.6. Method Validation

Method validation was conducted according to the validation scheme proposed in previous work [20,24]. The analytical parameters evaluated for each BP were: linearity, trueness expressed as mean percent recovery (RE, %), intra-day precision (repeatability) and inter-day precision (intermediate precision) as percent relative standard deviation (RDS, %), limits of detection (LODs) and limits of quantification (LOQs). Plant-based beverage samples analyzed to verify that they were not contaminated by BPs were used as blank samples and for method specificity evaluation. As shown in Figure 3, no interfering peak was detected in the diagnostic area of the -MRM ion chromatogram of the blank matrix, proving that the method had a good specificity.

The linearity of the detector response was assessed for each BP at 7 concentration levels: 0.78, 1.56, 3.12, 6.25, 12.5, 25.0 and 50.0 ng/mL. Two types of calibration curves, solvent calibration and matched-matrix calibration curve, were constructed to evaluate the matrix effect (ME). Matched-matrix calibration solutions were prepared by adding known volumes of the BP working solutions to blank sample extracts. To assess whether the matrix significantly affected the peak area and thus the sensitivity of the method, ME was evaluated by applying the following Equation (1):ME (%) = B/A × 100(1)
A = peak area obtained by adding the analytes to solvent consisted of ultrapure water and MeOH in the ratio of 50/50 (*v*/*v*).B = peak area obtained by spiking plant-based beverage extracts with the analytes.

A value of 100% indicates that the response in the solvent (ultrapure water and MeOH in the ratio of 50/50 (*v*/*v*) and matrix is the same, a value of > 100% indicates an increased ionization, and a value < 100% indicates a suppressed ionization [25]. Very little ionization suppression was observed for BPA, BPB, and BPS, with values of 95.0%, 97.0%, and 96.5%, respectively. This confirms that ME was not significant. However, matched-matrix calibrations were preferred and employed for quantification. Trueness was calculated as RE (%), while precision was calculated as RSD (%) for both repeatability and intermediate precision. LOQs were obtained by fortifying blank samples at decreasing concentrations of each analyte; the lowest concentration showing a chromatographic peak with a signal-to-noise ratio ≥ 10 was assumed to be an LOQ value. The LOQ value for all BPs was found to be 0.78 ng/mL. The LOD of each BP was calculated according the following equation: LOQ = 3.3 × LOD and was thus estimated to be 0.24 ng/mL for all BPs. In each working session, a process blank and a matrix blank were also analyzed for confirming the absence of background contamination by BPs. The process blank consisted only of the solvent used during the extraction procedure, passed onto the same cartridges used for the extraction of the analytes from the matrix.

## 3. Results and Discussion

### 3.1. Sample Preparation and LC/ESI-QqQ MS/MS

Sample pretreatment is an important step in isolating targeted compounds from a bulk food matrix and avoiding any interference in the final detection and quantification of analytes [26]. To obtain the best results in terms of interfering compound reduction and trueness, Liquid-Liquid (LL) and SPE extraction techniques were tested. In the former case, the type and amount of extraction solvent were modified and tested. For SPE, different stationary phases and types of solvents were considered. Considering the above factors, SPE on Strata X-Pro cartridges produced the best results and was therefore chosen for sample preparation. A Luna Omega Polar C18 has been chosen for chromatographic separation, having balanced retention of polar and hydrophobic compounds and the proper selectivity to ensure suitable separation of target compounds. In preliminary studies, different mobile phases (data not shown) and a linear gradient of ultrapure water and MeOH, both supplemented with 0.01% acetic acid, were tested. In particular, different amounts of acid additives were tried to achieve the best chromatographic efficiency. However, the addition of formic acid or a high amount of acetic acid (0.1%) exerted unwanted ion suppression in the negative ion mode, reducing the sensitivity of the analysis, while the addition of 0.01% acetic acid proved to be the best compromise, ensuring good chromatographic separation and a sensitive mass-spectrometric response. The gradient elution program was also optimized to achieve the best analytical performance in the shortest analysis time. To maximize the ionization efficiency under LC conditions and the mass-spectrometric response, full-scan mass spectra were obtained by injecting the analytes at flow and tuning the source parameters. Product ion spectra were acquired to select a quantifier ion (Q) and a qualifier ion (q) for each analyte to enable unambiguous determination of target compounds in the matrix of interest. Collision energies were selected by product ion and MRM experiment, also taking advantage of the automatic tool (Optimizer) provided by mass spectrometer manufacturer.

### 3.2. Method Validation

The developed method was validated to verify its applicability in routine analysis of plant-based beverage samples and to ensure the reliability of the results. The matrix-matched calibration method was applied to assess the linearity of the method. Correlation coefficient (R^2^) values in the range 0.992–0.998 were achieved for all three BPs, demonstrating the good linearity of the method. Two concentration levels of the three analytes, 10 ng/mL and 25 ng/mL were chosen for evaluating trueness and precision (intra- and inter-day, respectively) of the method. Two working sessions for each spiking level were conducted on different days using a total of eight blank samples fortified at the two selected validation levels and extracted. As shown in Table 3, satisfactory values were obtained for both the trueness and precision of the method at each concentration level and for all BPs. Actually, the mean percentage recovery was between 85.3% and 98.0% at 10 ng/mL (*n* = 8) and between 78.0% and 105.0% at 25 ng/mL (*n* = 8). RDS (%) values in the range of 6.4%-13.2% and between 8.1% and 14.1% were obtained for repeatability (RDS_r_) and intermediate precision (RDS_R_), respectively. An LOQ value of 0.78 ng/mL was established for all analytes while the LOD value was 0.26 ng/mL. As shown in our previous articles, LOQ values may vary depending on the analyte and the matrix [20,27]. We optimized the chromatographic method for the matrix under investigation and obtained comparable LOQs for all three analytes. Furthermore, the run time and, consequently, the amount of solvent required for the analysis are consistent with the applicability of the method to routine analyses.

### 3.3. BP Contamination in Plant-Based Beverages

The validated method was applied to the routine analysis of 34 samples of plant-based beverages bought in supermarkets in Naples and its province (Italy) between January and March 2021. Eleven different brands were selected (B1-B11) considering their commercial availability and their distribution in the Italian market. The samples included five different tastes: soya (*n* = 9); oats (*n* = 8); rice (*n* = 8); almond (*n* = 7); coconut (*n* = 2). All samples were packaged in tetra pack (27 in 1 L packages and 7 in 500 mL packages) and the expiration date did not exceed one year [20]. Table 4 summarizes the results of our investigation. The presence of BPA was detected in 32% of the samples analyzed above the LOQ in the range 1.0–18.70 ng/mL, while BPB was found in only one sample (3% with a concentration of 5.17 ng/mL). No samples were found to be contaminated with BPS. The detected compounds satisfied the requirements for concordance of retention times and ion ratios with standards in fortified blank samples.

An example of a plant-based beverage sample contaminated with BPA is shown in Figure 4. In the diagnostic area of the -MRM ion chromatogram of the considered sample, quantitative and qualitative transitions related to BPA can be observed (see middle inset), but not those related to BPS (left inset) and BPB (right inset).

To investigate the possible correlation of BPA levels with brand and taste, we used a one-way analysis of variance (ANOVA). All samples were grouped according to the taste (soya, coconut almond, oats, and rice) and the different brands of the plant-based beverages. In both cases, we obtained a *p*-value > 0.05 (0.084 and 0.063, respectively), which indicates that differences in concentration are not to be considered significant.

### 3.4. Chronic Risk Assessment

In this study, chronic dietary exposure to BPs through the consumption of plant-based beverages has been analyzed using a specific tool proposed by EFSA called RACE [28]. Calculations were performed using the median BPA levels obtained from the analysis of 34 real plant-based beverage samples by LC-MS/MS detection. The RACE tool was only used for BPA, as this is the BP mainly detected in the food matrix selected. The tool uses food consumption information from EFSA’s Comprehensive European Food Consumption Database to provide estimates of chronic exposure to individual foods, and risk assessment was obtained by comparing these exposure data with an appropriate toxicological reference point, namely the TDI. The results for long-term chronic exposure using the current TDI value (4 µg/kg bw/day) showed values of less than 1% indicating the absence of risk for all population groups (<100%). However, EFSA’s Panel on Food Contact Materials, Enzymes and Processing Aids (CEP), recently proposed a re-evaluation of the public health risks specifically associated with the presence of BPA in food. The CEP stated that exposure to BPA through food has increased in all age groups and therefore proposed to lower the TDI value to 0.04 ng/kg bw/day. EFSA is currently evaluating the proposal and will deliver its opinion by the end of 2022. Therefore, the risk assessment was also calculated using the proposed new TDI value (0.04 ng/kg bw/day). In this case, the results obtained are far above 100%, indicating that there would be a strong risk associated with the intake of BPs through the consumption of plant-based beverages if the proposed TDI value is accepted.

## 4. Conclusions

The results of our survey confirm the need to remain ever vigilant about the possible contamination of food by BPs. Our analysis shows that a food can be contaminated with BP even if it is packaged in a material that, like Tetra Pack^TM^, should not contain BP in its composition. BPA was detected in 32% of the plant-based beverage samples. The levels of BPA detected are not alarming when considering the current TDI value. However, the presence of BPA, even at low concentrations, can be harmful to human health due to so-called mixture effects related to the introduction of the same and/or different types of contaminants from different sources into the human body. Furthermore, if EFSA’s proposal to lower the current TDI value turns out to be correct, further efforts by the industry would be required to reduce the use of BPA and its congeners as much as possible to avoid risks to human health.

## Figures and Tables

**Figure 1 foods-11-03853-f001:**
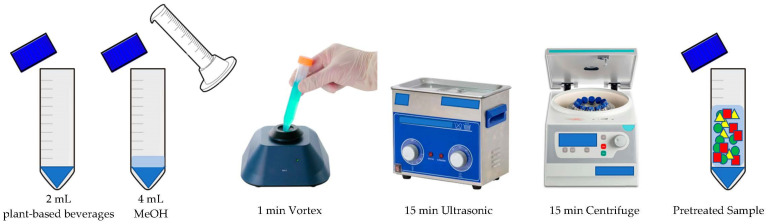
Drawing of sample pretreatment process.

**Figure 2 foods-11-03853-f002:**
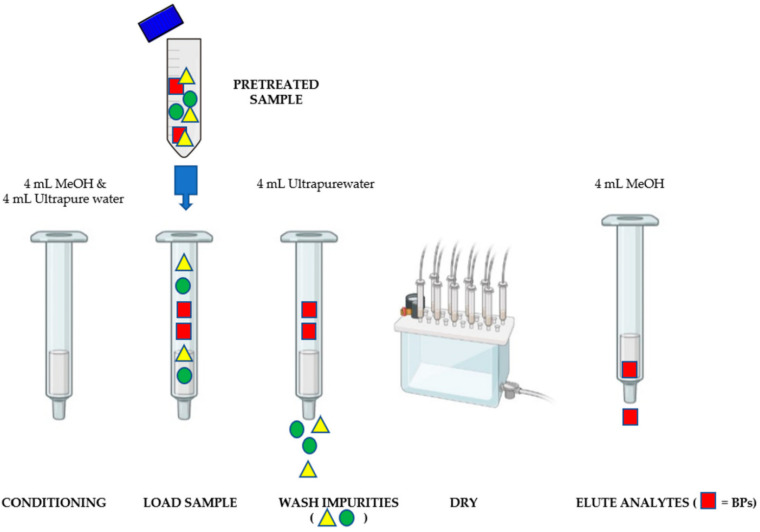
Illustrated scheme of the solid-phase extraction process.

**Figure 3 foods-11-03853-f003:**
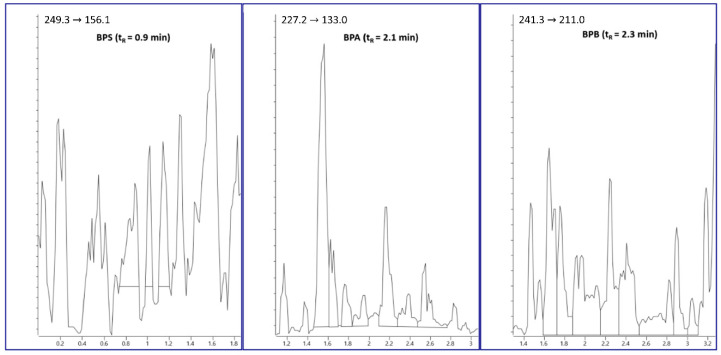
The -MRM ion chromatogram of the blank matrix.

**Figure 4 foods-11-03853-f004:**
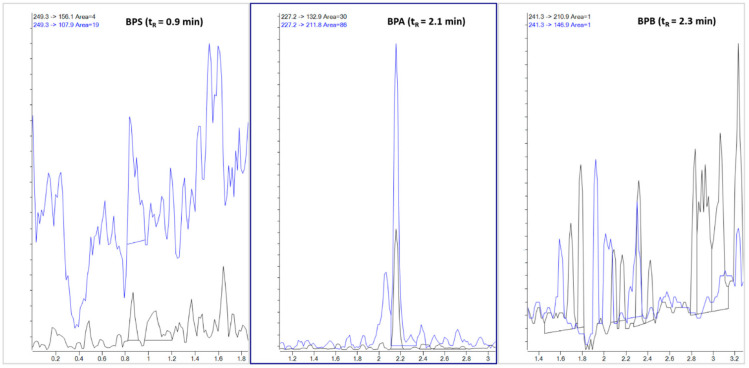
The -MRM ion chromatogram of a plant- based beverage sample contaminated with 2.6 ng/mL BPA. For each analyte, blue lines indicate the qualifier ions and black lines indicate the quantifier ions.

**Table 1 foods-11-03853-t001:** Gradient chromatographic elution optimized to separate BPs.

Time	Ultrapure Water with 0.01% Acetic Acid	MeOH Water with 0.01% Acetic Acid
0.0	60.00	40.00
0.5	60.00	40.00
3.0	5.0	95.00
4.0	5.0	95.00

**Table 2 foods-11-03853-t002:** UHPLC-MS/MS Quantifier and Qualifier transitions, Collision energy and Fragmentor used for BPA, BPB and BPS.

Production	Q1 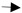 Q3*m*/*z*	Collision Energy	Fragmentor
BPA-Q	227.2 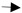 133.0	−20	162
BPA-q	227.2 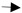 211.8	−28	162
BPB-Q	241.3 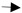 211.0	−40	110
BPB-q	241.3 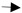 147.0	−45	110
BPS-Q	249.3 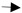 156.1	−30	130
BPS-q	249.3 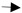 108.1	−30	130

**Table 3 foods-11-03853-t003:** Mean percentage recovery, repeatability, and intermediate precision for BPA, BPB and BPS at validation spiking levels.

Compound	Spiking Level (ng/mL)	Mean Percentage Recovery (%) ^1^	Repeatability (RSD_r_, %) ^2^	Intermediate Precision (RSD_R_, %) ^1^
BPA	10.0	98.0	11.0	11.1
	25.0	105.0	8.7	8.8
BPB	10.0 25.0	98.0 101.3	6.4 12.1	8.1 12.3
BPS	10.0 25.0	85.3 78.0	13.2 10.2	14.1 10.3

^1^ (*n* = 8). ^2^ (*n* = 4).

**Table 4 foods-11-03853-t004:** Bisphenol contamination level in 34 plant-based beverages collected from Italian market.

Sample	Taste	Brand *	BPA (ng/mL)	BPB (ng/mL)	BPS (ng/mL)
1	Almond	B1	<LOQ	<LOD	<LOD
2		B2	1.15	<LOD	<LOD
3		B3	<LOQ	<LOD	<LOD
4		B5	7.25	<LOD	<LOD
5		B6	1.14	<LOD	<LOD
6		B7	<LOD	<LOD	<LOD
7		B10	2.6	<LOD	<LOD
8	Oats	B1	3.75	<LOD	<LOD
9		B2	<LOQ	<LOD	<LOD
10		B3	<LOD	<LOD	<LOD
11		B4	<LOD	<LOD	<LOD
12		B5	18.17	<LOD	<LOD
13		B6	<LOD	<LOD	<LOD
14		B8	1.00	<LOD	<LOD
15		B9	<LOD	<LOD	<LOD
16	Rice	B2	1.50	<LOD	<LOD
17		B3	<LOD	<LOD	<LOD
18		B4	<LOD	<LOD	<LOD
19		B7	<LOD	<LOD	<LOD
20		B8	<LOD	<LOD	<LOD
21		B9	<LOQ	<LOD	<LOD
22		B10	<LOQ	<LOD	<LOD
23		B11	1.85	<LOD	<LOD
24	Soya	B1	<LOD	5.17	<LOD
25		B2	<LOD	<LOD	<LOD
26		B3	<LOD	<LOD	<LOD
27		B4	<LOD	<LOD	<LOD
28		B6	<LOQ	<LOD	<LOD
29		B7	<LOQ	<LOD	<LOD
30		B8	<LOD	<LOD	<LOD
31		B9	2.37	<LOD	<LOD
32		B10	<LOD	<LOD	<LOD
33	Coconut	B1	<LOD	<LOD	<LOD
34		B7	3.7	<LOD	<LOD

* For privacy reasons, the brand names are generically indicated by the letter B (Brand) followed by a number from 1 to 11, each indicating a different brand.

## Data Availability

The data presented in this study are available on request from the corresponding author.

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
