# Peer review of "Quantitative Determination of Bisphenol A and Its Congeners in Plant-Based Beverages by Liquid Chromatography Coupled to Tandem Mass Spectrometry"

_foods, 2022, doi:10.3390/foods11233853_

Round 1

Reviewer 1 Report

The manuscript presents a method of simultaneous determination of three bisphenols (BPA, BPB and BPS) by LC-ESI-QTRAP-MS/MS in plant-based beverages.

Authors applied (with some minor modifications) the earlier described method of endocrine-disrupting chemicals analysis (https://doi.org/10.1080/19440049.2022.2104933 - not cited, https://doi.org/10.1080/19440049.2019.1657967  - cited) for the evaluation of three bisphenols in a new matrix - plant-based beverages. It is important to monitor concentrations of such components in available food products due to their negative influence on human health.

The reviewed manuscript is well-written in English and well-prepared; the methods and the discussion are consequent. However, some shortcomings are noticed.

I would suggest the following revisions to be undertaken:

Introduction. Some discussion on the selected for the analysis chromatographic method should be provided in light of other available procedures.

Page 5, Figure 3 is not readable.

Composition of the mobile phases. The authors depicted a concentration of the mobile phase in several places in the manuscript, but it is still not determined why 0.01% of acetic acid is used. It should be emphasized if it works as a negative ionic agent or has some other function in this procedure. 

Page 7, line 256, the authors claim to optimize the analytical procedure, but any statistical design of experiment methods is described. It would be more informative if the authors would inform the reader it was one of a time “optimization” or maybe a more advanced procedure was undertaken.

Page 8, line 316, the authors claim that analysis of obtained results was conducted using the ANOVA, but neither the obtained result is presented in the manuscript nor in supporting information.

Page 9, Figure 4, there is no description of the meaning of blue and the black line. Without such information, the reader can not interpret the data depicted in this Figure.

Author Response

The manuscript presents a method of simultaneous determination of three bisphenols (BPA, BPB and BPS) by LC-ESI-QTRAP-MS/MS in plant-based beverages.

Authors applied (with some minor modifications) the earlier described method of endocrine-disrupting chemicals analysis (https://doi.org/10.1080/19440049.2022.2104933 - not cited, https://doi.org/10.1080/19440049.2019.1657967  - cited) for the evaluation of three bisphenols in a new matrix - plant-based beverages. It is important to monitor concentrations of such components in available food products due to their negative influence on human health.

The reviewed manuscript is well-written in English and well-prepared; the methods and the discussion are consequent. However, some shortcomings are noticed.

I would suggest the following revisions to be undertaken:

Introduction. Some discussion on the selected for the analysis chromatographic method should be provided in light of other available procedures.

We thank the reviewer for your suggestion. We have added a comment at page 7 lines 296-301

Page 5, Figure 3 is not readable.

In accordance with reviewer’s comment, we have changed relocated figure 3

Composition of the mobile phases. The authors depicted a concentration of the mobile phase in several places in the manuscript, but it is still not determined why 0.01% of acetic acid is used. It should be emphasized if it works as a negative ionic agent or has some other function in this procedure. 

In accordance with reviewer’s comment, we have added a comment on page 7 Lines 265-272

Page 7, line 256, the authors claim to optimize the analytical procedure, but any statistical design of experiment methods is described. It would be more informative if the authors would inform the reader it was one of a time “optimization” or maybe a more advanced procedure was undertaken.

We thank the reviewer for your suggestion. We have modified the sentence (page 7 Lines 258-263) and we hope we have satisfied the reviewer's request

Page 8, line 316, the authors claim that analysis of obtained results was conducted using the ANOVA, but neither the obtained result is presented in the manuscript nor in supporting information.

In accordance with reviewer’s comment, we have reported p-values. We hope we have satisfied the reviewer's request.

Page 9, Figure 4, there is no description of the meaning of blue and the black line. Without such information, the reader can not interpret the data depicted in this Figure.

In accordance with reviewer’s comment, we have modified the caption of figure 4.

Reviewer 2 Report

The manuscript entitled: Quantitative Determination of BPA and its Congeners in plant-based beverages by Liquid Chromatography Coupled to Tandem Mass Spectrometry is an interesting and novel work that could act as a guideline for determinations of BPA in food products. I have appended my comments in the attached PDF. Authors should pay attention to minor grammar and spell check.

Author Response

 We have reported our answers in the attached PDF
